# Genome-Wide Identification and Expression Analyses of Odorant-Binding Proteins in Hoverfly *Eupeodes corollae*

**DOI:** 10.3390/ijms26188956

**Published:** 2025-09-14

**Authors:** He Yuan, Huiru Jia, Xianyong Zhou, Hui Li, Chao Wu, Kongming Wu

**Affiliations:** 1State Key Laboratory for Biology of Plant Diseases and Insect Pests, Institute of Plant Protection, Chinese Academy of Agricultural Sciences, Beijing 100193, China; yuanhe1001@126.com (H.Y.); lihuilh521@163.com (H.L.); 2Xianghu Lab, Hangzhou 311258, China; jhuiru@163.com (H.J.); ZhouXY160721@163.com (X.Z.); 3Shenzhen Branch, Guangdong Laboratory of Lingnan Modern Agriculture, Genome Analysis Laboratory of the Ministry of Agriculture and Rural Affairs, Agricultural Genomics Institute at Shenzhen, Chinese Academy of Agricultural Sciences, Shenzhen 518000, China; wuchao@caas.cn

**Keywords:** odorant-binding proteins, *Eupeodes corollae*, conserved structures, evolutionary analysis, transcriptome, expression patterns

## Abstract

Chemosensory systems are fundamental for insects to regulate behaviors such as prey detection, oviposition, and pollination. Despite their importance, the molecular mechanisms underlying chemosensation remain poorly understood in many insect groups. Hoverflies (Syrphidae), whose larvae are efficient aphid predators and adults act as pollinators, represent a functionally important but understudied lineage. Building on the genome of *Eupeodes corollae* that we recently published, we selected this dominant and widespread species as a representative model and performed a genome-wide identification and analysis of odorant-binding proteins (OBPs) to provide a molecular foundation for understanding chemosensory recognition mechanisms. Accordingly, a total of 47 OBPs were identified and classified into Classic, Minus-C, and Plus-C subfamilies, with conserved motifs and structural features observed within each group. Next, phylogenetic analysis revealed that several *EcorOBPs* are homologous to functionally characterized OBPs in other Diptera, suggesting conserved evolutionary roles. Moreover, chromosomal mapping showed that Minus-C *EcorOBPs* cluster on chromosome 2, and Ka/Ks analysis indicated strong purifying selection, reflecting evolutionary stability. In addition, synteny analysis demonstrated that *E. corollae* shares more collinear OBP gene pairs with predatory hoverflies (*Episyrphus balteatus* and *Scaeva pyrastri*) than with the saprophagous species *Eristalis tenax*, consistent with ecological divergence. Finally, transcriptomic profiling revealed tissue-specific expression patterns, including antennal-biased *EcorOBP1* linked to olfaction and reproductive tissue-biased *EcorOBP11* linked to reproduction, highlighting candidate genes for functional studies. Together, these findings provide a comprehensive characterization of OBPs in *E. corollae* and offer molecular insights into chemosensory mechanisms that support both pest control and pollination services.

## 1. Introduction

Olfaction is one of the most important sensory modalities in insects, enabling them to locate food, identify mates, select oviposition sites, and avoid natural enemies. These behaviors are largely mediated by the detection of chemical cues from the environment, and understanding the molecular basis of insect olfaction is therefore essential for elucidating their ecological adaptations and interactions. Chemosensory proteins play central roles in this process, among which odorant-binding proteins (OBPs) are particularly important because they transport hydrophobic odor molecules through the aqueous sensillar lymph to the olfactory receptors [1,2,3].

Among chemosensory proteins, odorant-binding proteins (OBPs) play a pivotal role. OBPs are small (ca. 120–150 amino acids), water-soluble proteins with a hydrophobic pocket that transports odorant molecules to olfactory receptors [1,2,3]. They are widespread across insect orders, including Lepidoptera, Diptera, Coleoptera, and Hemiptera [4,5,6,7,8], with considerable variability in gene number among species—from fewer than ten in some parasitoids to more than one hundred in mosquitoes [7]. Advances in genome and transcriptome sequencing have greatly facilitated the identification of OBP repertoires. Based on conserved cysteine patterns, OBPs are classified into Classic, Minus-C, Plus-C, Dimer, and Atypical groups [9,10,11,12,13]. Such structural variation allows OBPs to bind diverse ligands and thereby mediate a wide spectrum of physiological and ecological functions.

In model Diptera such as *Drosophila melanogaster* and mosquitoes, OBPs have been functionally characterized in processes including host seeking, oviposition, and insecticide resistance [14,15,16,17,18,19,20,21,22,23]. In contrast, studies on other dipteran groups remain limited, and the molecular basis of olfaction in hoverflies (Diptera: Syrphidae) is still poorly understood. Hoverflies are of particular interest because their larvae are voracious aphid predators, while adults serve as important pollinators [24,25]. These dual ecological roles depend strongly on chemical communication, yet only fragmentary information is available on their OBP repertoires. Previous work has identified OBPs in *Eupeodes corollae* and *Episyrphus balteatus* [26,27,28,29], but the reported number of OBPs in *E. corollae* (28) was markedly smaller than in *E. balteatus* (45), raising questions about annotation completeness and highlighting the need for genome-based re-evaluation.

Building on the genome of *E. corollae* that we recently published, this study performed a genome-wide identification and comparative analysis of OBPs in this species. We examined their phylogenetic relationships, conserved motifs and domains, gene structures, and evolutionary patterns. We further compared collinear OBPs with other hoverfly species and analyzed their tissue-specific expression profiles using transcriptomic data. These analyses provide the first comprehensive characterization of OBPs in *E. corollae*, establishing a molecular foundation for future functional studies and offering new insights into the chemosensory mechanisms that underpin the ecological services of hoverflies.

## 2. Results

### 2.1. Identification of OBP Genes in E. corollae

This study used transcriptomic data from different tissues of adults *E. corollae* and annotated 47 OBPs, in which 25 new OBPs had never been described before. Then, we amplified the cDNA sequences of the remaining 25 OBP genes using PCR. We successfully amplified 25 OBPs, which were numbered 5, 14, 19, 21, 23, and 35–54 in the present study (Table 1). *EcorOBP48* had the longest Open reading frame (ORF), which was 1209 bp. The remaining 46 OBP genes in *E. corollae* contained ORFs ranging from 351 bp to 759 bp. The identified 25 OBPs sequences were deposited in GenBank under accession numbers PQ284629–PQ284652, and PQ846003.

Physicochemical properties: MW, isoelectric points, and signal peptides were identified and are listed in Table 1. Sequence analysis categorized 25 *EcorOBPs* into three subgroups: classic, plus-C, and minus-C OBPs, based on the number and pattern of cysteines. Eighteen OBPs (EcorOBP5, 14, 19, 21, 23, 35–38, 40, 42–44, 47–48, and 52–54) have six conserved Cys residues in the classic subgroups. *EcorOBP46,* with eight conserved Cys, was classified as plus-C OBPs, and EcorOBP49, 50, and 51, with four conserved Cys, as minus-OBPs. The remaining three OBPs (EcorOBP39, 41, and 45) did not have the conserved six or four Cys, but, according to the conserved C2 and C5, these three OBPs were found to be classic OBPs (Table 1, Figure 1).

### 2.2. Motif Compositions and Gene Structure of EcorOBP Genes

A maximum likelihood tree was constructed using the amino acid sequences of 47 *EcorOBPs* in *E. corollae*. Plus-C OBPs were found to be clustered into one clade, including *EcorOBP2*, *3*, *4*, and *46*. Similar motif compositions and conserved domains were observed in the four plus-C OBPs (Figure 2). Motif 5 was present in the four plus-C OBPs and absent in the other *EcorOBPs* except for *EcorOBP45*. No conserved domains were found in the four OBPs. Notably, *EcorOBP45* was clustered together with the four plus-C OBPs, displaying similar conserved motif compositions and domains. Conversely, sequence alignments revealed that *EcorOBP45* exhibited only four conserved cysteines, inconsistent with the signature of plus-C OBPs (eight conserved cysteines and one conserved proline).

For minus-C OBPs, seven *EcorOBPs* were clustered into a large clade, exhibiting similar conserved domains and intron–exon distribution. Each gene displayed 3–5 motifs. All seven minus-C OBPs displayed motifs 1 and 2. Minus-C OBPs exhibited typical PBP-GOBP subfamily structures and contained one intron (Figure 2).

The largest number of members (36/47) belonged to classic OBPs with a distinct number of conserved motifs and intron–exon distribution. Classic OBPs contained 2–6 motifs. Most OBPs (24/36) displayed motifs 4, 3, 1, and 2. *EcorOBP21* and *23* contained motif 6, motifs 8 and 10 were observed in *EcorOBP5* and *14*, and motif 9 was seen in *EcorOBP12* and *15*. Most members of classic OBPs (31/36) contained one conserved domain, GOBP, according to the conserved domain analysis. The “PhBP” domain was found in *EcorOBP52*. *EcorOBP48* and *19* share the domain “PBP-GOBP superfamily.” Intron–exon organization demonstrated that classic OBPs contained 1–5 introns (*EcorOBP20*), varying in size (Figure 2). Six OBPs, *EcorOBP8*, *9*, *12*, *16*, *42*, and *44*, displayed relatively longer introns. However, the exon sizes of *EcorOBPs* were relatively conserved.

### 2.3. Evolutionary Analyses of OBP Family

Phylogenetic clustering of genes suggested their close relationships. To infer the functions of OBPs in *E. corollae*, we compared OBPs in *E. corollae* with seven important insect species by phylogenetic analysis. The phylogenetic tree clustered OBP family into several clear clades, revealing 21 homologous subgroups between *E. corollae* (*EcorOBP1*, *2–5*, *9–14*, *15–18*, *20*, *22*, *28*, *30–31*, *38*, *40–42*, *44*, *46*, *51*) and *Bactrocera dorsalis* or *D. melanogaster* (i.e., *LUSH*, OBP8a, OBP19a, OBP19b, OBP19c, OBP28a, OBP44a, OBP47b, OBP50e, OBP56a, OBP56g, OBP59a, OBP69a, OBP73a, OBP83a, OBP83b, OBP83ef, OBP84a, OBP99b, OBP99c, OBP99d) (Figure 3). OBPs homologs in one subgroup might evolve from a common ancestor and have close relationships.

### 2.4. Chromosomal Locations

Analysis of the chromosomal distribution of *EcorOBP* genes found that 47 *EcorOBPs* genes were distributed across three of four chromosomes based on the *E. corollae* genome assembly (Figure 4). Moreover, 31 *EcorOBPs*, 15 *EcorOBPs*, and one gene (*EcorOBP9*) were distributed in chromosomes 1, 2, and 3, respectively. No OBPs were found on chromosome 4. Classic OBPs were dispersed on chromosomes 1, 2, and 3. Plus-c OBPs were situated on chromosome 1, while minus-c OBPs were closely spaced apart on chromosome 2, which were clustered in the interval of 143 kb. Several OBPs clusters were also observed (Figure 4). For example, OBP22, OBP26, OBP28, OBP31, and OBP35-OBP39 were clustered in the interval of 71 kb. OBP2 and OBP3 were clustered in the interval of 154 bp. However, the OBP gene clusters showed low amino acid sequence identity, suggesting they were not a recent gene duplication event.

### 2.5. Evolution Analysis of OBP Orthologs

The Ka, Ks, and Ka/Ks values were calculated based on gene sequences from *E. corollae* and *E. balteatus* to explore the selection pressure on *EcorOBP* genes. Synteny analysis of *E. corollae* and *E. balteatus* was carried out to compare homologous OBP genes, identifying gene pairs. The Ka/Ks values of all OBP genes were less than one, ranging from 0 to 0.6, indicating that genes were subjected to purifying selection (Table 2).

### 2.6. Synteny Analysis of OBP Genes in E. corollae and Other Hoverfly Species

To better understand the evolution of the OBPs gene family in *E. corollae*, we compared the genome sequences of *E. corollae* with other hoverflies species, including two predatory hoverflies, *Scaeva pyrastri* and *E. balteatus*, and one saprophagous hoverfly, *Eristalis tenax*. The synteny analysis demonstrated that *E. corollae* and *E. balteatus* shared 31 orthologous OBP genes, *E. corollae* and *S. pyrastri* shared 25 collinear OBP genes, and *E. corollae* and *E. tenax* shared 15 OBPs (Figure 5, Appendix A), indicating that there are more orthologous genes between species with close relationships. *E. corollae* contained 23 syntenic orthologs as compared to the two predatory hoverflies, including *EcorOBP1*, *EcorOBP2*, *EcorOBP4*, *EcorOBP5*, *EcorOBP7*, *EcorOBP9*, *EcorOBP10*, *EcorOBP13*, *EcorOBP15*, *EcorOBP16*, *EcorOBP18*, *EcorOBP19*, *EcorOBP21*, *EcorOBP30*, *EcorOBP40*, *EcorOBP41*, *EcorOBP42*, *EcorOBP44*, *EcorOBP46*, *EcorOBP48*, *EcorOBP50*, *EcorOBP51,* and *EcorOBP53* (Appendix A), in which 14 orthologous genes were present in *E. corollae* when compared to the three hoverflies.

### 2.7. Expression Profiles of EcorOBPs

The different expression profiles may be attributed to the distinct roles of the genes. The expression patterns of 47 *EcorOBPs* genes based on the FPKM values in different tissues of *E. corollae* adults were illustrated through a heat map (Figure 6, Appendix A). Antennae serve as the primary olfactory organs in recognizing and perceiving chemical signals in the environment. Twelve *EcorOBPs* genes (*EcorOBP1*, *9*, *12*, *15–17*, *20*, *31*, *37*, *39*, *40*, and *46*) exhibited higher expression levels in the antennae of adult females and males compared to the other tissues. Eight OBPs (*EcorOBP1*, *9*, *12*, *16*, *17*, *20*, *40*, and *46*) were enriched explicitly in the antennae of both sexes. In insects, proboscises and legs are two important taste organs that may also contribute to gustation. The highest expression levels were displayed by the five OBPs (*EcorOBP4*, *23*, *34*, *44*, and *47*) in proboscises, followed by legs. *EcorOBP26* and *43* were exclusively enriched in proboscises. *EcorOBP7*, *18*, and *49* expression levels were higher in heads (without antennae or proboscises) than in the other tissues. *EcorOBP10*, *11*, *13*, *30*, *21*, *50,* and *51* displayed higher expression in the abdomens, where the reproductive organs were located, suggesting their physiological roles beyond chemoreception. *EcorOBP2* and *28* exhibited abundant expressions among the examined tissues, which might affect a variety of fundamental physiological processes.

## 3. Discussion

With the increasing availability of genomic data, genome-wide mining, identification, and characterization of novel genes have become more efficient, providing valuable resources for evolutionary and functional studies of OBPs. Genomic sequencing identified numerous OBP genes from more than 100 insect species [30]. The number of OBPs varies among species. For Diptera, the number of OBPs ranges from 41 to 62 in *Drosophila* species, exceeding 100 in several mosquitoes, such as *A. aegypti* (111) and *Culex quinquefasciatus* (109) [3,12,31]. This study performed a genome-scale identification of OBP genes (47) in *E. corollae* based on its reference genome, which was more than the number (28) identified by transcriptomes [26,28]. The number of OBPs was equivalent to that in hoverfly *E. balteatus* (44) and comparable to those in *Drosophila* [27].

The characteristic features of OBPs, including sequence models, conserved motifs and domains, and gene structures, were analyzed to further explore OBP genes in *E. corollae*. In total, 33 members of the OBP gene family displayed a typical six-cysteine pattern, referred to as classic OBPs. The remaining three OBPs (*EcorOBP39*, *41*, and *45*) also belong to classic OBPs according to the conserved C2 and C5. Moreover, 11 OBPs were classified into two subgroups: minus-C OBPs with four cysteines and plus-C OBPs with two additional cysteines. Regarding the same subfamily, motif compositions were comparable. Plus-C OBPs displayed motif 5, absent in other OBPs except for *EcorOBP45*. However, different classes presented different gene structures, implying their distinct evolution histories.

The evolution of OBPs in *E. corollae* was further examined using phylogenetic analysis, Ka/Ks calculation, and synteny relationships among different hoverfly species. Minus-C OBPs depicted a more conserved evolution than the two other classes. The members of the minus-C OBPs were clustered into one clade. Chromosome distribution found that the members were closely located on the same chromosome (Chr2), indicating close relationships of genes and their evolution from the same ancestral gene, consistent with the plus-C evolution in *D. melanogaster* [13]. Ka/Ks calculation demonstrated that OBP genes of *E. corollae* were subjected to purifying selection, consistent with prior findings in *E. balteatus* [27]. Collinearity relationships were analyzed to investigate gene evolution among different hoverfly species. A greater number of collinear gene pairs were identified between *E. corollae* and two predatory hoverflies, *E. balteatus* and *S. pyrastri*, than between *E. corollae* and the saprophagous hoverfly, *E. tenax*. The differences in the number of collinear gene pairs could be attributed to evolutionary divergence and phylogenetic relationships. We also observed that the homologous OBP gene pairs are distributed on different chromosomes among species, which might be caused by genome rearrangements, such as duplication and translocation, which are common in hoverflies and other insect species [32,33].

To a certain degree, the expression patterns of genes acquired through transcriptomic sequencing could be used to realize their functions [3,6]. OBP genes expressed in olfactory organs are involved in chemosensory functions [14,34]. OBPs in non-sensory tissues are essential in physiological processes such as oviposition, development, and insecticide [18,35,36]. This study revealed broad tissue expression patterns of OBPs in adults *E. corollae*. Nine OBPs that were specifically expressed in the antennae of *E. corollae* indicate their involvement in olfaction. Seven OBPs highly expressed in the abdomen that contains reproductive organs may be implicated in oviposition behavior [3,7]. Volatiles from flowering plants are important indicators of pollination behavior. This study found that eight *EcorOBPs*, which were highly expressed on taste organs (legs or mouthparts), might be involved in gustatory recognition and detection of plant nectars or pollens, like *AcerOBP15* and *PregOBP56a* [37,38]. Tissue-specific expression of OBPs is reported per each subfamily in *Manduca sexta* [9]. For example, classic OBPs were more restricted to olfactory tissues such as antennae, while plus-C and minus-C OBPs were expressed on many tissues. Similarly, the findings of this study revealed that 11 out of 12 highly expressed OBPs in antennae were classic OBPs, two of seven minus-C OBPs were highly expressed in adult heads, and five minus-C OBPs in abdomens. This study provides evidence for the functional characterization of the OBP gene family.

Volatile compounds are vital chemical cues in the plant–insect–natural enemy interactions, including plant, herbivore-induced plant, and insect volatiles [39]. The perception of chemical signals is crucial for insects in mating, oviposition, and locating prey and host plants. In many aphid species, EBF is an alarm pheromone acting as a warning signal, attracting natural enemies to forage, and serving as an oviposition stimulant for *E. corollae* [40,41]. Reportedly, some OBPs in aphids and aphid predators could bind to EBF, such as *ApisOBP3*, *ApisOBP7*, *HaxyOBP15*, *HvarOBP5*, and *MperOBP3*, indicating their important role in locating prey [14,42,43,44]. There is limited data on the function of *EcorOBPs* except for *EcorOBP15* [29]. This study offers cues for inferring the functions of other OBPs in *E. corollae* through phylogenetic analysis, synteny analysis, and expression patterns of OBPs in various tissues of adults. Among the homologous OBPs in *E. corollae* identified by phylogenetic analysis, *EcorOBP12*, *EcorOBP15*, *EcorOBP17*, *EcorOBP20,* and *EcorOBP40*, which were mainly expressed on the antennae of *E. corollae*, might be involved in pheromone and phytochemicals perception. Indeed, *EcorOBP15* was confirmed in mediating attraction to EBF produced by plants and aphids [29]. In addition, antennae-rich *EcorOBP1* clustered together with *DmelOBP59a*, which was involved in humidity sensing [45]. In addition, *EcorOBP11* clustered with *DmelOBP8a*, which was related to oviposition [46,47]. Considering that *EcorOBP11* was highly expressed in the abdomen of *E. corollae,* which houses reproductive organs, we speculated that *EcorOBP11* might be involved in the reproductive behavior of *E. corollae*. However, further experimental investigations are needed to explore *EcorOBPs*’ functions in the prey location, pollination, and reproductive behaviors of *E. corollae*.

## 4. Materials and Methods

### 4.1. Transcriptomic Analysis

A TransZol UP Plus RNA Kit (TransGen Biotech, Beijing, China) was used to extract total RNA from adult *E. corollae* following the recommended protocols. The RNA integrity was evaluated via gel electrophoresis, and RNA purity was evaluated via a spectrophotometer (Thermo Scientific, Waltham, MA, USA). Furthermore, cDNA was synthesized from 1 μg of total RNA by a HiScript Ⅱ 1st Strand cDNA Synthesis Kit (Vazyme, Nanjing, China) and used as a template for amplifying OBP genes.

For transcriptomic sequencing, cDNA libraries were constructed using high-quality RNA from each tissue and sequenced with a PE150 strategy employing the Illumina Novaseq 6000 system at Shanghai Majorbio Biotech Corporation (Shanghai, China). TopHat2 (v2.1.1) was used to map the clean reads to the *E. corollae* reference genome [48,49]. Cufflinks (v2.2.1) were used to assemble mapped reads into transcripts [50]. RSEM (RNA-Seq by Expectation-Maximization) (v1.3.3) was used to normalize the expression levels of transcripts to Fragments Per Kilobases per Millionreads (FPKM) [51]. DESeq2 (v1.10.1) [52] was used to obtain differentially expressed genes with a cut-off of |log_2_FC| ≥ 1 and padjust < 0.05. The data of different tissues transcriptomes of *E. corollae* in this study are available at NCBI as BioProject PRJNA1160340. The transcriptome sequencing data have been deposited in the NCBI SRA under accession number SRS22641170-SRS22641193.

### 4.2. Identification and Physicochemical Properties of OBPs

A prior study identified 46 OBP genes from *E. corollae* genome assembly [48]. In this study, the transcriptomic data from different tissues of *E. corollae* adults were blasted against the reference genome to improve or refine the OBP gene family in *E. corollae*. Then, we manually annotated all the OBPs in *E. corollae* using BlastP against the NCBI non-redundant (nr) protein sequence database with e-value cutoff of 10^−5^ and obtained the best hit.

Jia et al. [26] identified *EcorOBPs*, while other candidate genes encoding *EcorOBPs* were verified using PCR amplification. The specific primers of open reading frame sequences of OBPs are provided in Appendix A. The Super Pfx MasterMix (Cwbio, Taizhou, China) was used to carry out the PCR reaction under the following conditions: 98 °C for 3 min, 35 cycles of 98 °C for 10 s, 55 °C for 30 s, 72 °C for 30 s, and 72 °C for 10 min. The PCR products were purified, cloned into the pEasy-T3 vector (TransGen, Beijing, China), and sequenced.

The SignaIP 5.0 server (http://www.cbs.dtu.dk/services/SignalP/ (accessed on 20 August 2024)) was used to predict the putative N-terminal signal peptides and cleavage sites [53]. ExPASy ProtParam (https://web.expasy.org/protparam/ (accessed on 20 August 2024)), an online tool, was employed to compute the following parameters: molecular weights (MW) and theoretical pI. Multiple sequence alignments of OBP amino acid sequences without signal peptides were conducted using MUSCLE, which was implemented in MEGA 11.0 software. The results were edited using GeneDoc 2.7. The phylogenetic tree of *E. corollae* OBP genes was generated through MEGA 11.0 using the maximum likelihood approach [54]. The reliability of the tree was tested by bootstrap analysis with 1000 replications.

### 4.3. Gene Structure and Conserved Motif Analysis

MEME suite 5.5.6 (https://meme-suite.org/meme/tools/meme (accessed on 20 August 2024)) was employed to foretell the conserved motifs of *EcorOBPs*. The number of motifs was set to 10, with other parameters as default. NCBI conserved domain database (https://www.ncbi.nlm.nih.gov/Structure/bwrpsb/bwrpsb.cgi (accessed on 25 August 2024)) was searched to identify conserved structural domains. The exon–intron structure of the *EcorOBP* genes was acquired from its genome database. TBtools (v2.337) was used to visualize the results [55].

### 4.4. Evolutionary Analyses of OBP Genes

The amino acid sequences of predicted OBPs from *E. corollae* and two Dipteran speices (*D. melanogaster* and *B. dorsalis*) were aligned with ClustalW. Two OBPs (*ApisOBP1* and *ApisOBP2*) from the aphid (*A. pisum*) were used as outgroups. Then, a maximum likelihood tree was built using MEGA 11.0 with default settings. The unrooted tree was viewed and edited using the itol tool (v7) (http://itol.embl.de (accessed on 20 June 2025)).

### 4.5. Chromosomal Localization and Synteny Analysis

Genome-wide gene density in *E. corollae* was calculated using the genome database. All OBP genes were mapped to the *E. corollae* reference genome using BLAST v2.3.0+ searches to illustrate their physical locations on chromosomes. Chromosomal distribution and the relative position of the OBP gene family were visualized through TBtools (v2.337) [55]. The genomic information of three hoverfly species, including *S. pyrastri*, *E. balteatus*, and *E. tenax*, was acquired from NCBI (Appendix A) [32]. Gene synteny and collinearity between *E. corollae* and other hoverfly species were analyzed using MCScanX software [56]. Those homologous OBP genes between *E. corollae* and other species were represented in red. The mapping of comparable gene pairs was realized via TBtools (v2.337) [55].

### 4.6. Ka/Ks Calculation

The whole-genome synteny analysis above identified orthologous gene pairs between *E. corollae* and *E. balteatus*, two common predatory hoverflies. To explore the evolutionary dynamics of OBPs, Ka and Ks substitution rates and ratio (Ka/Ks) were calculated based on protein-coding sequences of the identified orthologous OBPs using the KaKs_Calculator 3.0 with NG model [57]. The input sequences were generated using ParaAT (v2.0) with default settings [58].

### 4.7. Insect Rearing and Tissue Collection

In 2015, adult *E. corollae* were collected from wheat fields in Langfang city, Hebei Province, China. The laboratory culture was established and reared at a 23 ± 1 °C temperature and a photoperiod consisting of 14 h of light and 10 h of darkness. Aphids on bean plantlets were fed to the larvae, and adult *E. corollae* were given pollen and 10% (*v*/*v*) honey–water solution. For tissue collection, adult females and males were split following eclosion.

Antennae were dissected from newly emerged adult females and males (n = 100), along with other tissues, including proboscises, heads (without antennae and proboscises), thoraxes, abdomens, legs, and wings (n = 30) of mixed sexes (male:female = 1:1), and collected for transcriptomic sequencing. Each tissue sample was subjected to three replicates. Liquid nitrogen was used to freeze the tissue samples immediately, which were then stored at –80 °C until RNA isolation. These tissue samples were prepared for subsequent transcriptome sequencing.

## 5. Conclusions

Genome-wide identification and refinement of OBP genes in *E. corollae* were conducted in this study. A total of 47 OBPs were identified and classified into three subgroups: classic, plus-C, and minus-C OBPs. OBP genes in each subgroup exhibited similar motif compositions, conserved domains, and gene structures. Chromosomal mapping and Ka/Ks analysis indicated the evolutionary stability of *EcorOBPs*. Synteny analysis among hoverflies revealed that *EcorOBPs* have undergone genetic divergence. Phylogenetic analysis revealed several homologous OBPs to those in other Diptera species, and tissue-specific expression patterns of *EcorOBPs* indicated their functional diversity, which provided candidate genes for functional studies. For example, *EcorOBP1* might be involved in humidity sensing, and *EcorOBP11* might be involved in oviposition. This study offers a solid foundation to explore the molecular mechanisms and potential functions of OBPs in the prey location and pollination of *E. corollae*, aiding in the pest control of natural enemies and plant biodiversity conservation.

## Figures and Tables

**Figure 1 ijms-26-08956-f001:**
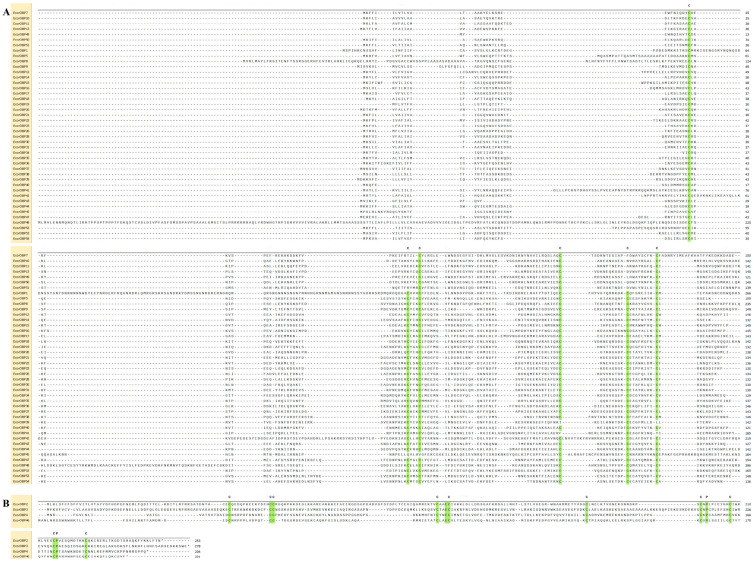
Alignment of full-length amino acid sequences of OBPs in *E. corollae*. The conserved Cys residues (C1–C6) are indicated in green color. (**A**) Alignment of classic and minus-C *EcorOBPs*. (**B**) Plus-C *EcorOBPs*.

**Figure 2 ijms-26-08956-f002:**
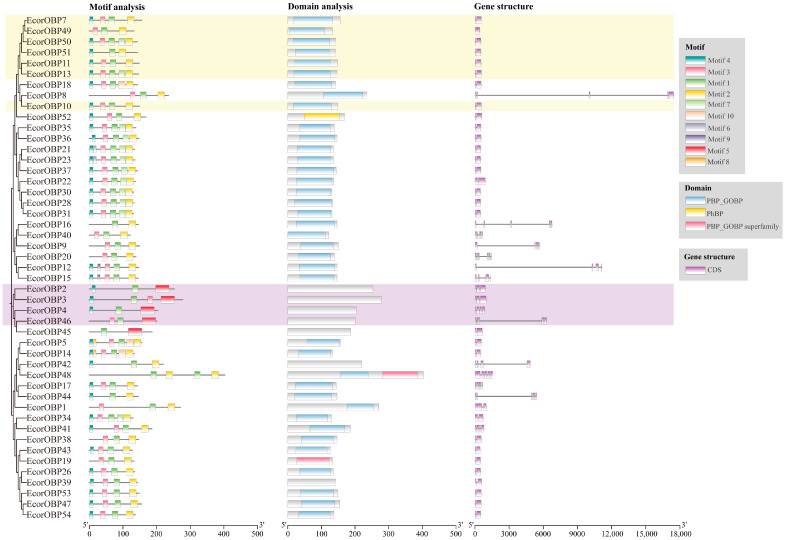
Phylogenetic tree, motif, conserved domain, and gene structure of the *EcorOBP* family in *E. corollae*. The phylogenetic tree of the OBP proteins was constructed using the maximum likelihood (ML) method by MEGA11 with 1000 bootstrap replicates. The yellow background represents minus-C OBPs, and the purple background represents plus-C OBPs.

**Figure 3 ijms-26-08956-f003:**
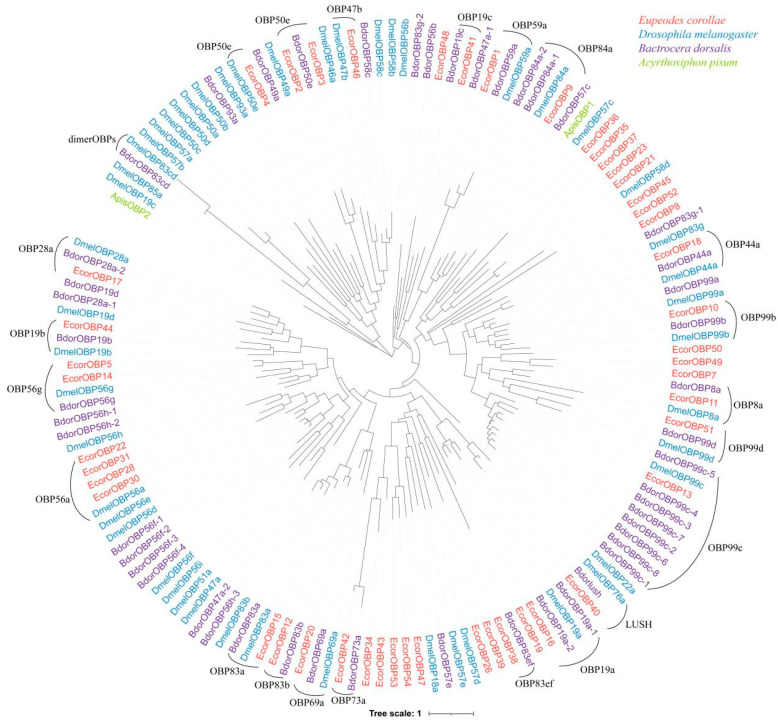
Maximum likelihood tree of putative OBPs in *E. corollae* with other insect species. Ecor: *E. corollae*; Dmel: *D. melanogaster*; Bdor: *B. dorsalis*; Apis: *Acyrthosiphon pisum*.

**Figure 4 ijms-26-08956-f004:**
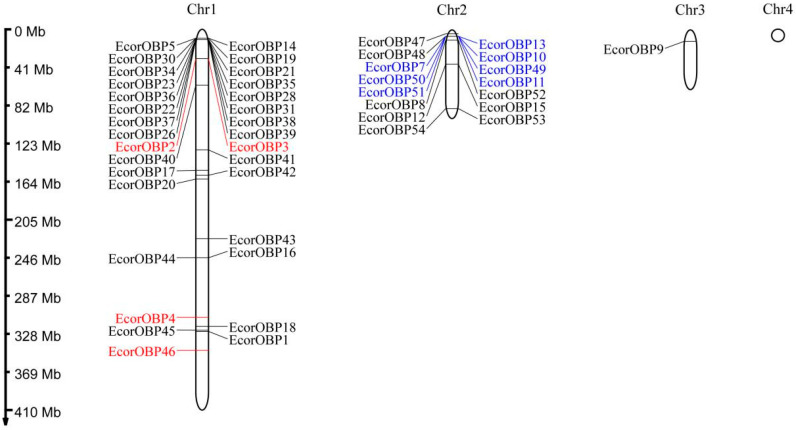
The genomic positions of *EcorOBPs* genes. OBPs in red indicate plus-C OBPs. OBPs in blue indicate minus-C OBPs, and black represents classic OBPs.

**Figure 5 ijms-26-08956-f005:**
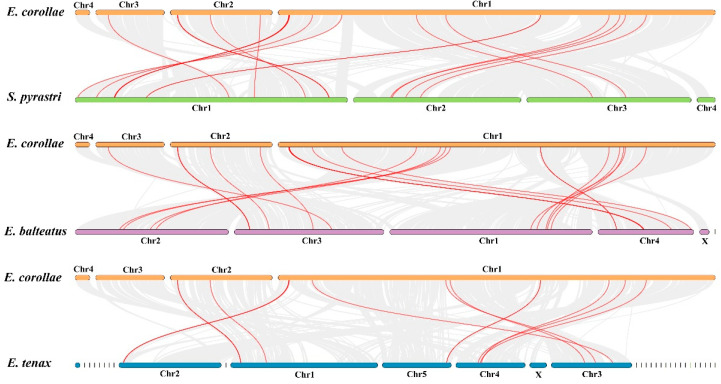
Collinearity analysis of OBP proteins in *E. corollae* among species. The species include *S. pyrastri*, *E. balteatus*, and *E. tenax*. The gray line indicates the collinear block between hoverfly genomes, and the red line indicates the homologous OBP gene pairs.

**Figure 6 ijms-26-08956-f006:**
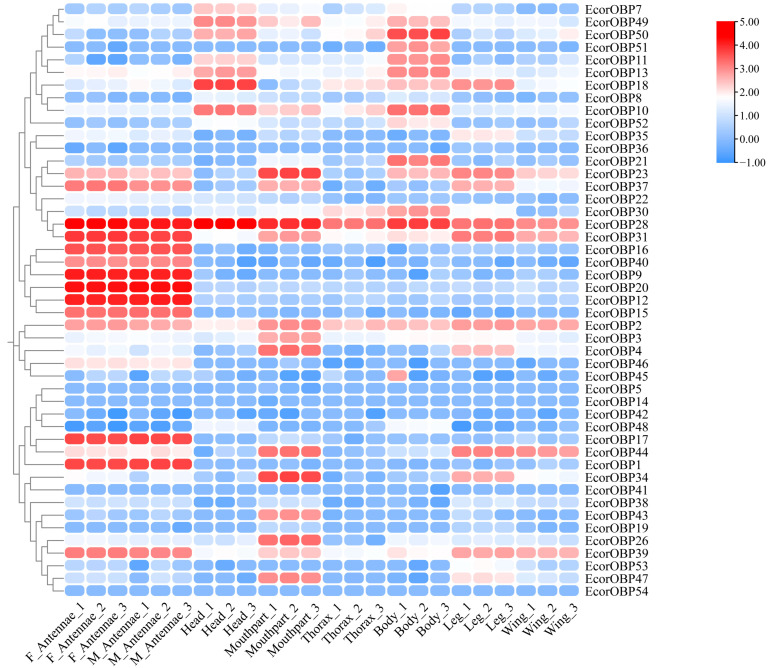
Expression heatmap of *EcorOBPs* in different tissues of adults in *E. corollae*. FAn, female antennae; MAn, male antennae; Head, heads without antennae or proboscises. Blue color indicates down-regulated expression of genes. Red color indicates up-regulated expression of genes; the hotter (redder) the color, the higher the gene expression level.

**Table 1 ijms-26-08956-t001:** List of 47 OBPs identified in *E. corollae* in this study.

	Accession Number	ORF (bp)	SP (aa)	MM (kDa)	pI	Class	Best Blastp Hit		
							Gene Annotation	*E*-Value	Identity %
*EcorOBP1*	MT585316	351	1–19	14.02	7.66	Classic	*EcorOBP1*	0.0	99.63
*EcorOBP2*	MT585317	759	1–18	28.94	7.86	Plus-C	*EcorOBP2*	0.0	96.83
*EcorOBP3*	MT585318	570	––	21.14	6.6	Plus-C	*EcorOBP3*	6 × 10^−131^	98.19
*EcorOBP4*	MT585319	612	1–23	22.61	6.23	Plus-C	*EcorOBP4*	6 × 10^−148^	99.51
*EcorOBP5* ^n^	PQ284629	468	1–19	17.18	8.45	Classic	uncharacterized protein LOC129943204 [*Eupeodes corollae*]	1 × 10^−106^	97.42
*EcorOBP7*	MT585321	468	1–16	18.44	5.59	Minus-C	*EcorOBP7*	2 × 10^−108^	98.71
*EcorOBP8*	MT585322	456	1–19	17.46	5.73	Classic	*EcorOBP8*	6 × 10^−92^	92.31
*EcorOBP9*	MT585323	450	1–18	16.53	4.88	Classic	*EcorOBP9*	3 × 10^−104^	99.33
*EcorOBP10*	MT585324	447	1–16	16.94	6.09	Minus-C	*EcorOBP10*	8 × 10^−105^	100
*EcorOBP11*	MT585325	444	1–18	16.40	5.07	Minus-C	*EcorOBP11*	7 × 10^−102^	100
*EcorOBP12*	MT585326	441	1–20	16.49	5.52	Classic	*EcorOBP12*	7 × 10^−104^	100
*EcorOBP13*	MT585327	441	1–16	16.61	6.32	Minus-C	*EcorOBP13*	3 × 10^−101^	100
*EcorOBP14* ^n^	PQ284630	399	1–19	14.69	5.67	Classic	general odorant-binding protein 57c [*Eupeodes corollae*]	1 × 10^−91^	100
*EcorOBP15*	MT585328	438	1–21	16.78	5.44	Classic	*EcorOBP15*	2 × 10^−102^	99.31
*EcorOBP16*	MT585329	438	1–23	16.28	7.5	Classic	*EcorOBP16*	3 × 10^−99^	97.93
*EcorOBP17*	MT585330	432	1–24	15.40	5.2	Classic	*EcorOBP17*	3 × 10^−97^	100
*EcorOBP18*	MT585331	429	1–18	16.66	9.11	Classic	*EcorOBP18*	6 × 10^−98^	100
*EcorOBP19* ^n^	PQ284651	399	1–21	15.21	4.71	Classic	uncharacterized protein LOC129919207 [*Episyrphus balteatus*]	7 × 10^−48^	58.02
*EcorOBP20*	MT585332	411	1–21	15.80	5.13	Classic	*EcorOBP20*	1 × 10^−96^	99.28
*EcorOBP21* ^n^	PQ284631	405	1–20	14.87	4.04	Classic	general odorant-binding protein 56h-like [*Eupeodes corollae*]	2 × 10^−89^	99.25
*EcorOBP22*	MT585333	411	1–20	15.37	5.53	Classic	*EcorOBP22*	2 × 10^−93^	100
*EcorOBP23* ^n^	PQ284632	408	1–20	15.13	4.14	Classic	uncharacterized protein LOC129939419 [*Eupeodes corollae*]	8 × 10^−91^	100
*EcorOBP26*	MT585336	405	1–19	15.12	5.54	Classic	*EcorOBP26*	9 × 10^−86^	94.78
*EcorOBP28*	MT585338	399	1–18	14.71	5.83	Classic	*EcorOBP28*	5 × 10^−89^	99.24
*EcorOBP30*	MT585340	393	1–20	14.63	5.12	Classic	*EcorOBP30*	1 × 10^−86^	98.46
*EcorOBP31*	MT585341	393	1–18	14.79	5.96	Classic	*EcorOBP31*	1 × 10^−89^	99.23
*EcorOBP34*	MT585344	390	1–18	14.59	4.51	Classic	*EcorOBP34*	2 × 10^−86^	99.22
*EcorOBP35* ^n^	PQ284633	414	1–21	15.97	8.12	Classic	general odorant-binding protein 56a-like [*Eupeodes corollae*]	3 × 10^−39^	44.85
*EcorOBP36* ^n^	PQ284634	441	1–23	16.89	4.98	Classic	uncharacterized protein LOC129940025 [*Eupeodes corollae*]	8 × 10^−96^	96.58
*EcorOBP37* ^n^	PQ284635	432	1–25	16.46	5.13	Classic	uncharacterized protein LOC129939358 [*Eupeodes corollae*]	7 × 10^−97^	99.30
*EcorOBP38* ^n^	PQ284636	441	1–19	16.50	4.65	Classic	uncharacterized protein LOC129940488 [*Eupeodes corollae*]	3 × 10^−101^	100
*EcorOBP39* ^n^	PQ284637	429	1–22	16.76	5.03	Classic	uncharacterized protein LOC129938798 [*Eupeodes corollae*]	1 × 10^−90^	97.18
*EcorOBP40* ^n^	PQ284638	366	5′missing	13.71	8.18	Classic	general odorant-binding protein lush [*Eupeodes corollae*]	4 × 10^−84^	100
*EcorOBP41* ^n^	PQ284639	558	1–20	21.05	7.59	Classic	uncharacterized protein LOC129946411 [*Eupeodes corollae*]	8 × 10^−128^	97.30
*EcorOBP42* ^n^	PQ284640	660	1–19	25.27	5.91	Classic	general odorant-binding protein 70 [*Eupeodes corollae*]	1 × 10^−162^	100
*EcorOBP43* ^n^	PQ284641	384	1–19	14.68	4.91	Classic	general odorant-binding protein 56d-like [*Eupeodes corollae*]	3 × 10^−87^	98.43
*EcorOBP44* ^n^	PQ284642	438	1–20	16.40	4.55	Classic	uncharacterized protein LOC129939463 [*Eupeodes corollae*]	3 × 10^−102^	100
*EcorOBP45* ^n^	PQ284652	561	––	21.07	9.34	Classic	uncharacterized protein LOC129952676 [*Eupeodes corollae*]	4 × 10^−133^	98.39
*EcorOBP46* ^n^	PQ284643	603	1–29	22.47	6.70	Plus-C	uncharacterized protein LOC129952205 [*Eupeodes corollae*]	2 × 10^−147^	99.50
*EcorOBP47* ^n^	PQ284644	465	1–20	17.84	4.45	Classic	uncharacterized protein LOC129939928 [*Eupeodes corollae*]	5 × 10^−106^	97.40
*EcorOBP48* ^n^	PQ846003	1209	––	45.96	7.47	Classic	uncharacterized protein LOC129942694 [*Eupeodes corollae*]	0.0	98.76
*EcorOBP49* ^n^	PQ284645	399	1–19	15.94	5.62	Minus-C	*EcorOBP6*	1 × 10^−80^	88.64
*EcorOBP50* ^n^	PQ284646	429	1–15	16.61	6.38	Minus-C	general odorant-binding protein 99a-like [*Eupeodes corollae*]	4 × 10^−95^	99.30
*EcorOBP51* ^n^	PQ284647	429	1–19	16.21	4.65	Minus-C	uncharacterized protein LOC129942711 [*Eupeodes corollae*]	7 × 10^−96^	97.18
*EcorOBP52* ^n^	PQ284648	504	1–19	19.25	5.18	Classic	general odorant-binding protein 99a-like [*Eupeodes corollae*]	5 × 10^−116^	97.61
*EcorOBP53* ^n^	PQ284649	447	1–22	16.76	5.22	Classic	uncharacterized protein LOC129940557 [*Eupeodes corollae*]	9 × 10^−102^	99.32
*EcorOBP54* ^n^	PQ284650	411	1–19	15.63	5.01	Classic	uncharacterized protein LOC129941715 [*Eupeodes corollae*]	1 × 10^−93^	99.26

^n^ OBPs were newly identified and successfully amplified in this study.

**Table 2 ijms-26-08956-t002:** Ka, Ks, and Ka/Ks values of orthologous OBP genes from *E. corollae* and *E. balteatus.*

Gene Pair Compared	Ka	Ks	Pairwise Ka/Ks
*EcorOBP1*-*EbalOBP2*	0.10131	1.05918	0.09565
*EcorOBP2*-*EbalOBP1*	0.24240	1.19253	0.20327
*EcorOBP4*-*EbalOBP3*	0.23683	0.95730	0.24740
*EcorOBP7*-*EbalOBP18*	0.34741	1.07076	0.32445
*EcorOBP8*-*EbalOBP26*	0.61546	1.44500	0.42592
*EcorOBP9*-*EbalOBP13*	0.09568	0.68299	0.14009
*EcorOBP10*-*EbalOBP12*	0.20224	0.70834	0.28552
*EcorOBP11*-*EbalOBP14*	0.29320	0.80486	0.36429
*EcorOBP12*-*EbalOBP20*	0.00877	0.26438	0.03320
*EcorOBP13*-*EbalOBP21*	0.21247	1.11337	0.19083
*EcorOBP15*-*EbalOBP23*	0.02958	0.29025	0.10191
*EcorOBP16*-*EbalOBP22*	0.03274	1.46490	0.02235
*EcorOBP17*-*EbalOBP24*	0.19724	0.90357	0.21829
*EcorOBP18*-*EbalOBP25*	0.03152	0.62774	0.05021
*EcorOBP20*-*EbalOBP31*	0.06263	0.90399	0.06928
*EcorOBP23*-*EbalOBP32*	0.31054	0.70558	0.44013
*EcorOBP37*-*EbalOBP29*	0.26758	1.12227	0.23843
*EcorOBP40*-*EbalOBP17*	0.09451	1.08560	0.08705
*EcorOBP44*-*EbalOBP19*	0.19349	0.95005	0.20366
*EcorOBP46*-*EbalOBP4*	0.12787	0.86276	0.14822
*EcorOBP48*-*EbalOBP26*	0.89590	2.20138	0.40697
*EcorOBP49*-*EbalOBP8*	0.34413	0.85256	0.40364
*EcorOBP52*-*EbalOBP45*	0.66046	1.06255	0.62157

## Data Availability

Data is provided within the manuscript or Appendix A. The data of different tissues transcriptomes of *E. corollae* in this study are available at the website (https://www.ncbi.nlm.nih.gov/bioproject/PRJNA1160340, accessed on 13 September 2024).

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
