# Peer review of "Genome-Wide Identification and Expression Analyses of Odorant-Binding Proteins in Hoverfly Eupeodes corollae"

_ijms, 2025, doi:10.3390/ijms26188956_

Round 1
Reviewer 1 Report
Comments and Suggestions for Authors
Question 1:
In the collinearity analysis of OBP proteins in E. corollae among species (Fig. 6), the homologous OBP gene pairs (indicated by the red line) are observed to be located on different chromosomes. How can this phenomenon be explained from the perspective of genetic evolution?
Question 2:
In line 248-251, a greater number of collinear gene pairs were identified between E. corollae and two predatory hoverflies, E. balteatus and S. pyrastri, than between E. corollae and the saprophagous hoverfly, E. tenax. The authors proposed that the differences in the number of collinear gene pairs could be attributed to their distinct feeding habits. Could these findings suggest that the hoverfly has incorporated certain genes from its prey—or even from the host plants of its prey—into its own genome through trophic chains?
Author Response
Question 1: In the collinearity analysis of OBP proteins in E. corollae among species (Fig. 6), the homologous OBP gene pairs (indicated by the red line) are observed to be located on different chromosomes. How can this phenomenon be explained from the perspective of genetic evolution?
Response: Revised. A brief explanation has been added in the Discussion section (Lines 254-257). Genome rearrangements such as duplication and translocation often take place during species evolution. Collinearity analysis between species could reflect their genomic conservation and evolutionary relationships. The relative positions and arrangement of genes on certain regions or chromosomes, and the number or size of collinear regions could help to speculate the differentiation time between species. This phenomenon is common in hoverflies and other insect species (Ji et al., 2024; Zhu et al., 2024). Additionally, collinear regions often contain genes and regulatory elements that are crucial to life activities. By analyzing these regions, functional conserved genes and genomic structural variation that exist among species can be identified.
Reference:
Ji, J., Gao, Y., Xu, C., Zhang, K., Li, D., Li, B., Chen, L., Gao, M., Huangfu, N., Elumalai, P., Gao, X., Zhu, X., Wang, L., Luo, J., & Cui, J. (2024). Chromosome-level genome assembly of marmalade hoverfly Episyrphus balteatus (Diptera: Syrphidae). Scientific data, 11(1), 844.
Zhu, X., Yang, Y., Li, Q., Li, J., Du, L., Zhou, Y., Jin, H., Song, L., Chen, Q., & Ren, B. (2024). An expanded odorant-binding protein mediates host cue detection in the parasitic wasp Baryscapus dioryctriae basis of the chromosome-level genome assembly analysis. BMC biology, 22(1), 196.
Question 2: In line 248-251, a greater number of collinear gene pairs were identified between E. corollae and two predatory hoverflies, E. balteatus and S. pyrastri, than between E. corollae and the saprophagous hoverfly, E. tenax. The authors proposed that the differences in the number of collinear gene pairs could be attributed to their distinct feeding habits. Could these findings suggest that the hoverfly has incorporated certain genes from its prey—or even from the host plants of its prey—into its own genome through trophic chains?
Response: Revised (Line 253-254). We appreciate this interesting comment. Horizontal gene transfer occurs in many insect species from their host plants, which enables them to have better adaptability. For example, the whitefly has acquired the plant-derived phenolic glucoside malonyltransferase gene BtPMaT1, which enables whiteflies to neutralize phenolic glucosides (Xia et al., 2021). At present, there is no evidence for horizontal gene transfer through trophic chains in hoverflies. The observed differences are more likely due to evolutionary divergence and phylogenetic relationships. Further research on horizontal genes transfer in hoverflies will be a promising study.
Reference:
Xia, J., Guo, Z., Yang, Z., Han, H., Wang, S., Xu, H., Yang, X., Yang, F., Wu, Q., Xie, W., Zhou, X., Dermauw, W., Turlings, T. C. J., & Zhang, Y. (2021). Whitefly hijacks a plant detoxification gene that neutralizes plant toxins. Cell, 184(7), 1693-1705.e17.
Reviewer 2 Report
Comments and Suggestions for Authors
This study systematically identified 47 OBPs in Eupeodes corollae. Integrating evolutionary and expression analyses, it unveiled the potential roles of OBPs in predation and pollination, providing crucial molecular insights for pest biological control and plant biodiversity conservation. The methods were rigorous, the data reliable, and the findings hold significant application prospects. But some specific corrections are necessary:
Line 92: The Latin species names in the table should be italicized.
Line 93: In Figure A, the sequence signatures of several Classic and Minus-C OBPs are not apparent, which appears inconsistent with the descriptions in Table 1. For instance, EcorOBP5 is listed as a Classic member in the table, yet Figure A does not clearly display the six conserved cysteine residues. We recommend that the authors explicitly state the criteria used to classify OBPs into the Classic, Plus-C and Minus-C groups.
Line 121: The authors should provide a more thorough investigation into why these five OBPs lack identifiable conserved domains and why EcorOBP45 clusters with the four Plus-C OBPs. Additionally, the scale bars in the figure overlap; please adjust the layout accordingly.
Line 169: The gene names in the figure are densely labeled; please rearrange or resize them for clarity.
Lines 340–344: The authors used KaKs_Calculator to estimate Ka and Ks, but the specific model and parameter settings were not reported. Please supply these details.
Author Response
Reviewer #2:
This study systematically identified 47 OBPs in Eupeodes corollae. Integrating evolutionary and expression analyses, it unveiled the potential roles of OBPs in predation and pollination, providing crucial molecular insights for pest biological control and plant biodiversity conservation. The methods were rigorous, the data reliable, and the findings hold significant application prospects. But some specific corrections are necessary:
Question 1: Line 92: The Latin species names in the table should be italicized.
Response: We thank the reviewer for the careful observation. All Latin species names in Table 1 have been thoroughly checked and italicized in the revised manuscript.
Question 2: Line 93: In Figure A, the sequence signatures of several Classic and Minus-C OBPs are not apparent, which appears inconsistent with the descriptions in Table 1. For instance, EcorOBP5 is listed as a Classic member in the table, yet Figure A does not clearly display the six conserved cysteine residues. We recommend that the authors explicitly state the criteria used to classify OBPs into the Classic, Plus-C and Minus-C groups.
Response: We appreciate the reviewer’s suggestion, and have added the classification criteria in the Results section in the revised manuscript (Lines 109-114) as well as to revise the Figure 1. Based on the structural characteristics and phylogenetic relationships, OBPs are assigned into five subgroups: classical OBPs (six conserved cysteines), minus-C OBP (four conserved cysteines), plus-C OBP (eight conserved cysteines), atypical OBP (9-10 conserved cysteines and a long C-terminus), and dimer OBP (12 conserved cysteines) (Gu et al., 2011; Jia et al., 2020; Zhou et al., 2004). In our study, sequence analysis categorized 25 EcorOBPs into three subgroups: classic, plus-C, and minus-C OBPs, based on the number and pattern of cysteines. 18 OBPs (EcorOBP5, 14, 19, 21, 23, 35-38, 40, 42-44, 47-48 and 52-54) have six conserved Cys residues in the classic subgroups. EcorOBP46 with eight conserved Cys was classified as plus-C OBPs, while EcorOBP49, 50, and 51 with four conserved Cys as minus-OBPs. The remaining 3 OBPs (EcorOBP39, 41 and 45) did not have the conserved six or four Cys, but according to the conserved C2 and C5, these 3 OBPs were found to be classic OBPs, which was also observed in other species, such as Spodoptera exempta (Dong et al., 2021).
Reference:
Dong, Y., Li, T., Liu, J., Sun, M., Chen, X., Liu, Y., & Xu, P. (2021). Sex- and stage-dependent expression patterns of odorant-binding and chemosensory protein genes in Spodoptera exempta. PeerJ, 9, e12132.
Gu, S. H., Wang, S. P., Zhang, X. Y., Wu, K. M., Guo, Y. Y., Zhou, J. J., & Zhang, Y. J. (2011). Identification and tissue distribution of odorant binding protein genes in the lucerne plant bug Adelphocoris lineolatus (Goeze). Insect biochemistry and molecular biology, 41(4), 254-263.
Jia, H. R., Niu, L. L., Sun, Y. F., Liu, Y. Q., & Wu, K. M. (2020). Odorant Binding Proteins and Chemosensory Proteins in Episyrphus balteatus (Diptera: Syrphidae): Molecular Cloning, Expression Profiling, and Gene Evolution. Journal of insect science (Online), 20(4), 15.
Zhou, J. J., Huang, W., Zhang, G. A., Pickett, J. A., & Field, L. M. (2004). "Plus-C" odorant-binding protein genes in two Drosophila species and the malaria mosquito Anopheles gambiae. Gene, 327(1), 117-129.
Question 3: Line 121: The authors should provide a more thorough investigation into why these five OBPs lack identifiable conserved domains and why EcorOBP45 clusters with the four Plus-C OBPs. Additionally, the scale bars in the figure overlap; please adjust the layout accordingly.
Response: We thank the reviewer for this helpful comment. The five OBPs without identifiable conserved domains likely represent highly divergent members, as they still contain signal peptides and cysteine frameworks typical of OBPs. EcorOBP45 is classified as a Classic OBP based on six conserved cysteines, but it clustered with Plus-C OBPs due to similarities in motif composition and sequence features. The figure has also been redrawn with separated scale bars and improved layout.
Question 4: Line 169: The gene names in the figure are densely labeled; please rearrange or resize them for clarity.
Response: Accepted. The figure has been adjusted to improve readability.
Question 5: Lines 340–344: The authors used KaKs_Calculator to estimate Ka and Ks, but the specific model and parameter settings were not reported. Please supply these details.
Response: Done. We have supplemented the details according to the suggestion in the Method section and the corresponding revisions have been added in the revised manuscript (Lines 359, 362-364).
Reviewer 3 Report
Comments and Suggestions for Authors
Manuscript review report
Journal: International Journal of Molecular sciences (IJMS)
Manuscript number: ijms-3819957
Title MS: Genome-wide identification and expression analyses of odorant-binding proteins in hoverfly Eupeodes corollae
Section: Molecular Biology
Yuan et al studies well the transcriptome analysis of hover fly. Before accepting the manuscript authors need to revise these sections Abstract, introduction and discussion. The results are presented well. Species name should be italic throughout the MS including in the references.
Title: Ok
Abstract: Suggestion genes nae should be italic throughout MS, which stage is efficient predators must mention and maintain the end to first sequence. Phylogenetic tree results absent in abstract. In the keywords add transcriptome in sequence manner. Some functional or identified genes must presence in the abstract.
Introduction:
2nd last sentence should be move to the first paragraph or intro start with insect (hoverfly) or Transcriptome then OBP
Results: All the cited references statements should move to discussion section.
Discussion: What kind of mining?
Materials and methods: Start from RNA extraction or subtitle 4.7 move to first paraph
Conclusion: Needs to be revise to the points.

Author Response
Question 1: Yuan et al studies well the transcriptome analysis of hover fly. Before accepting the manuscript authors need to revise these sections Abstract, introduction and discussion. The results are presented well. Species name should be italic throughout the MS including in the references.
Response: We apologize for this oversight. All species names in the text, figures, tables, and references have been carefully checked and italicized in the revised manuscript.
Question 2: Abstract: Suggestion genes nae should be italic throughout MS, which stage is efficient predators must mention and maintain the end to first sequence. Phylogenetic tree results absent in abstract. In the keywords add transcriptome in sequence manner. Some functional or identified genes must presence in the abstract.
Response: We thank the reviewer for the valuable comments. Gene names have been italicized in the revised manuscript. The Abstract has been revised and optimized according to these suggestions (Lines 22-45).
Question 3: Introduction: 2nd last sentence should be move to the first paragraph or intro start with insect (hoverfly) or Transcriptome then OBP.
Response: We appreciate this suggestion. The introduction has been adjusted and reorganized accordingly (Lines 62-97).
Question 4: Results: All the cited references statements should move to discussion section.
Response: Revised. The references statements in the Results section have been moved to the Discussion section as suggested.
Question 5: Discussion: What kind of mining?
Response: We thank the reviewer for pointing out this ambiguity. The term “mining” in the original manuscript has been revised to “genome-wide mining” to clarify that it refers to the bioinformatic identification of OBP genes from genomic data (Lines 223-225).
Question 6: Materials and methods: Start from RNA extraction or subtitle 4.7 move to first paraph.
Response: Accepted. The section 4.7 in Materials and methods has been reorganized according to the suggestion.
Question 7: Conclusion: Needs to be revise to the points.
Response: Revised. The conclusion has been further refined and made more concise (Lines 379-386).
Reviewer 4 Report
Comments and Suggestions for Authors
In this study, the authors identified odorant-binding proteins (OBPs) and classified them in E. corollae. Phylogenetic analyses were performed based on motif composition, conserved domains, and gene structure of the OBPs. Additionally, they revealed that these genes are under purifying selection. Furthermore, through transcriptomic analysis, the authors found the tissue-specific expression patterns of OBPs. Collectively, these results provide valuable insights into the biological roles of OBPs in E. corollae. Nevertheless, the authors need to carefully review and revise this manuscript.
Major points:
- In this study, the authors performed several analyses based on limited data. Some of the conclusions are presented too definitively without further experimental validation. Therefore, it seems necessary to properly tone down these statements and carefully revise the sentences.
Minor points:
- Page 2, lines 77-79: “This study used transcriptomic data from different tissues of adults E. corollae and annotated 50 OBPs, updating the OBPs gene family of E. corollae.” However, Table 1 includes only 47 OBPs. Could the authors clarify why all 50 annotated OBPs were not presented in Table 1?
- Page 2, lines 81-82: “We successfully amplified 25 OBPs, which were numbered 5, 14, 19, 21, 23, 35-54 in present study” – But, Table 1 does not contain an entry for EcorOBP54.
- Table 1: For the reader’s convenience, please clearly mark which 25 OBPs were successfully amplified and which 28 OBPs newly identified.
- Table 1: There appear to be two entries labeled as EcorOBP31. Please correct this duplication.
- Figure 1: It is not easy to check the sequences in the black marks. If possible, please modify the figure to improve legibility.
- Please carefully check typos and grammatical errors. Ex, “Consistent with the OBPs distribution in Drosophila melanogaster [30-31].”
- Figure 4: In the result section, the authors describe the class and cluster of OBPs. To improve clarity for the reader, please add the class information along with EcorOBP number in the figure.
- Please include the version information of bioinformatics tools or software used in this study.
Author Response
Major points:
Question 1: In this study, the authors performed several analyses based on limited data. Some of the conclusions are presented too definitively without further experimental validation. Therefore, it seems necessary to properly tone down these statements and carefully revise the sentences.
Response: Thank you for highlighting this point. We carefully revised the statements in the revised manuscript (Lines 158-159, 267, 287, 289, 292).
Minor points:
Question 1: Page 2, lines 77-79: “This study used transcriptomic data from different tissues of adults E. corollae and annotated 50 OBPs, updating the OBPs gene family of E. corollae.” However, Table 1 includes only 47 OBPs. Could the authors clarify why all 50 annotated OBPs were not presented in Table 1?
Response: Revised. We annotated 50 OBPs based on transcriptomic data from different tissues of adults E. corollae, but we successfully amplified 47 OBPs by PCR amplification. The remaining 3 OBPs were not obtained by PCR. Thus we list these 47 OBPs in Table 1. To avoid any possible confusion, we have changed the number of OBPs identified to 47 in the revised manuscript (Line 101).
Question 2: Page 2, lines 81-82: “We successfully amplified 25 OBPs, which were numbered 5, 14, 19, 21, 23, 35-54 in present study” – But, Table 1 does not contain an entry for EcorOBP54.
Response: Revised. EcorOBP54 is listed in Table 1.
Question 3: Table 1: For the reader’s convenience, please clearly mark which 25 OBPs were successfully amplified and which 28 OBPs newly identified.
Response: Revised. These 25 OBPs that were successfully amplified were listed in Table 1, and labeled an ‘n’ at the top right corner of these 25 OBPs and made a note below the table (Line 116). The remaining 3 OBPs were not listed in Table 1.
Question 4: Table 1: There appear to be two entries labeled as EcorOBP31. Please correct this duplication.
Response: We apologize for this oversight. We have removed a duplicate of EcorOBP31 in Table 1.
Question 5: Figure 1: It is not easy to check the sequences in the black marks. If possible, please modify the figure to improve legibility.
Response: Revised. Figure 1 has been adjusted to improve legibility.
Question 6: Please carefully check typos and grammatical errors. Ex, “Consistent with the OBPs distribution in Drosophila melanogaster [30-31].”
Response: We thank the reviewer for pointing this out. We made careful revisions and improvements to the language and grammar in the revised manuscript.
Question 7: Figure 4: In the result section, the authors describe the class and cluster of OBPs. To improve clarity for the reader, please add the class information along with EcorOBP number in the figure.
Response: Revised. Based on the class information (classic, minus-C and plus-C OBPs), we labeled EcorOBPs with different colors in Figure 4. It was also indicated in the figure caption (Lines 175-176).
Question 8: Please include the version information of bioinformatics tools or software used in this study.
Response: Revised. We added the version information of bioinformatics tools or software used in the revised manuscript (Lines 305-307, 309, 341, 347, 352, 357, 362-363).
Reviewer 5 Report
Comments and Suggestions for Authors
Review of the article from Yuan et al., titled:
Genome-wide identification and expression analyses of odorant-binding proteins in hoverfly Eupeodes corollae
General comments:
In the manuscript authors reports genome-wide identification of odorant-binding proteins (OBPs) in Eupeodes corollae. Although it can be considered as an updated version of their previous study, I found this still interesting as it comprehensively updates the list of OBPs in this species with additional transcriptome based expression details! I am glad to find that the study has been conducted with sufficient attention to detail and results were presented appropriately. Here I review the overall quality of the manuscript and suggests improvements. First of all, the quality of figures (especially Fig. 1, 2 and 4) can be improved for clarity and readability. Second, the evolutionary analysis is not strong enough for the functional interpretations based on orthology. So, I suggest to avoid assumptions based on the current phylogenetic trees. Third, please pay more attention to include details of each steps in methods section. Finally, could you clearly mention the details of transcriptomes in NCBI database? Details can be found in the minor concerns.
Overall, the manuscript needs only minor improvements. However, I would like to point that the details are necessary.
Minor concerns:
L79: says 50 OBPs; but in overall authors report 47 OBPs. Please correct/describe the other 3 OBPs.
L82: Please check if EcorOBP48 is a dimer-OBP or not. DimerOBPs are reported in Drosophila melanogaster (for example: DmelOBP83cd) and even tetramer OBPs are found in beetles like Ips duplicatus. Additional insights will be interesting for readers.
Table 1: Italicize species names.
L97: Please remove phylogenetic analysis in title 2.2. there is no analysis, but only a representing phylogeny and always please mention the method used.
L137: by D. melanogaster > in D. melanogaster. Please check the use of "by" throughout the manuscript.
L139: Please note that, these are predictions. And with large evolutionary distance and low sequence we cannot conclude anything based on predictions. So add the element of potential/possible role in so and so. For example, instead of “play similar roles” use - could potentially involve in…
L149: Please remove- “share conserved functions”. Since it is not proven experimentally, add as could be involved in.
L153: Figure 3: Please add the method of phylogenetic analysis in title. Try to include dimerOBPs from Dmel. Do you have any outgroup? Add rooted or unrooted tree.
L219: what the color bar represents? Add a very detailed figure caption.
L294: Materials and methods: this section needs more clarity in terms of details.
L312: I understand that, MEGA 11.0 was used. But add all the details like, method of multiple sequence alignment, with or without signal peptides? If its ML tree, the bootstraps.
L315: why ML-tree in L315 and NJ-tree in L327?
L372: adding the tissue specific information (origin) of each transcriptome (in NCBI) would have been much more beneficial. Otherwise the work will not be much useful for future reference.
Author Response
General comments:
Question 1: First of all, the quality of figures (especially Fig. 1, 2 and 4) can be improved for clarity and readability.
Response: Done. The figures have been redrawn with higher resolution and clearer labeling to improve clarity and readability.
Question 2: Second, the evolutionary analysis is not strong enough for the functional interpretations based on orthology. So, I suggest to avoid assumptions based on the current phylogenetic trees.
Response: Revised. Functional interpretations have been revised with more cautious wording, avoiding overstatements based solely on phylogenetic trees.
Question 3: Third, please pay more attention to include details of each steps in methods section.
Response: Done. The Methods section has been revised and expanded with additional details to ensure clarity and reproducibility.
Question 4: Finally, could you clearly mention the details of transcriptomes in NCBI database?
Response: Revised. In the revised manuscript, we have clarified that the tissue samples described in Section 4.6 (antennae, proboscises, heads, thoraxes, abdomens, legs, and wings) were used for transcriptome sequencing (Lines 371-376). In Section 4.1, we also added the accession number of the corresponding NCBI SRA dataset (Lines 311-313).
Minor concerns:
Question 5: L79: says 50 OBPs; but in overall authors report 47 OBPs. Please correct/describe the other 3 OBPs.
Response: Corrected. Due to the other 3 OBPs were not amplified by PCR in this study, the number of OBPs has been unified to 47 (Line 101).
Question 6: L82: Please check if EcorOBP48 is a dimer-OBP or not. DimerOBPs are reported in Drosophila melanogaster (for example: DmelOBP83cd) and even tetramer OBPs are found in beetles like Ips duplicatus. Additional insights will be interesting for readers.
Response: We appreciate the reviewer’s insightful comment. We carefully re-examined the sequence of EcorOBP48. Although this gene shows an unusually long sequence compared with other OBPs, it possesses the six conserved cysteine residues that define Classic OBPs and lacks the two cysteine frameworks required for dimer-OBPs. We added the dimer-OBP of Dmel (DmelOBP83cd) and re-constructed the phylogenetic tree, and the result showed that EcorOBP48 and DmelOBP83cd clustered in two different branches (Figure 3). Therefore, EcorOBP48 should be retained in the Classic OBP subfamily rather than classified as a dimer.
Question 7: Table 1: Italicize species names.
Response: Revised. All Latin species names in Table 1 have been thoroughly checked and italicized in the revised manuscript.
Question 8: L97: Please remove phylogenetic analysis in title 2.2. there is no analysis, but only a representing phylogeny and always please mention the method used.
Response: Revised as suggested. The section title has been corrected and the method specified (Lines 121-122).
Question 9: L137: by D. melanogaster > in D. melanogaster. Please check the use of "by" throughout the manuscript.
Response: Done. The usage has been corrected throughout the manuscript.
Question 10: L139: Please note that, these are predictions. And with large evolutionary distance and low sequence we cannot conclude anything based on predictions. So add the element of potential/possible role in so and so. For example, instead of “play similar roles” use - could potentially involve in…
Response: Thank you for highlighting this point. We have carefully revised the statements throughout the manuscript as suggested.
Question 11: L149: Please remove- “share conserved functions”. Since it is not proven experimentally, add as could be involved in.
Response: Revised. We have removed this statement accordingly.
Question 12: L153: Figure 3: Please add the method of phylogenetic analysis in title. Try to include dimerOBPs from Dmel. Do you have any outgroup? Add rooted or unrooted tree.
Response: Revised. The method has been added in the title of Figure 3 (Line 161). A dimerOBP (DmelOBP83cd) from Dmel were included in the phylogenetic tree, and two OBPs (ApisOBP1 and ApisOBP2) from the aphid (Acyrthosiphon pisum) were used as outgroups (Lines 344-345). The unrooted tree was clarified in the revised manuscript (Lines 346).
Question 13: L219: what the color bar represents? Add a very detailed figure caption.
Response: Revised. A detailed explanation has been added to the figure caption (Lines 219-221).
Question 14: L294: Materials and methods: this section needs more clarity in terms of details.
Response: Revised. The section has been expanded with clearer and more detailed descriptions.
Question 15: L312: I understand that, MEGA 11.0 was used. But add all the details like, method of multiple sequence alignment, with or without signal peptides? If its ML tree, the bootstraps.
Response: Revised. The requested details have been added to the Methods section (Lines 329-331, 333-334).
Question 16: L315: why ML-tree in L315 and NJ-tree in L327?
Response: Revised. A maximum likelihood tree was used in Figure 2, and a neighbor-joining tree was used in Figure 3. To maintain consistency, we constructed the phylogenetic tree in Figure 2 and Figure 3 both using ML method in the revised manuscript (Lines 122, 147, 161, 333, 345).
Question 17: L372: adding the tissue specific information (origin) of each transcriptome (in NCBI) would have been much more beneficial. Otherwise the work will not be much useful for future reference.
Response: Revised. In the revised manuscript, we have clarified that the tissue samples described in Section 4.6 (antennae, proboscises, heads, thoraxes, abdomens, legs, and wings) were used for transcriptome sequencing (Lines 371-376).
Round 2
Reviewer 4 Report
Comments and Suggestions for Authors
The author has addressed all of the reviewer's points in the revised manuscript, and I have no further comments.